# Cementitious Insulated Drywall Panels Reinforced with Kraft-Paper Honeycomb Structures

**Sepideh Shahbazi, Nicholas Singer, Muslim Majeed, Miroslava Kavgic and Reza Foruzanmehr \***

Civil Engineering Department, University of Ottawa, 161 Louis Pasteur, Ottawa, ON K1N 6N5, Canada
\* Correspondence: reza.foruzan@uottawa.ca

**Abstract:** Standard building practices commonly use gypsum-based drywall panels on the interior wall and ceiling applications as a partition to protect the components of a wall assembly from moisture and fire to uphold the building code and ensure safety standards. Unfortunately, gypsum-based drywall panels have poor resistance to water and are susceptible to mold growth in humid climates. Furthermore, the accumulation of drywall in landfills can result in toxic leachate impacting the surrounding environment. A proposed solution to the pitfalls of gypsum-based drywall arises in its substitution with a new lightweight composite honeycomb sandwich panel. This study aimed to develop sandwich panels with improvements in flexural strength and thermal insulating properties through the combined use of cementitious binder mix and kraft-paper honeycomb structures. The proposed alternative is created by following standard practices outlined in ASTM C305 to create cement panels and experimenting with admixtures to improve the material performance in order to cater to a drywall panel application. The kraft-paper honeycomb structure is bonded to cured cementitious panels to create a composite "sandwich panel" assembly. The results indicate that the sample flexural strength performed well after 7 days and exhibited superior flexural strength at 28 days, while providing a substantial increase in R-value of 5.84 $m^2K/W$ when compared to gypsum-based panels, with an R-value of 5.41 $m^2K/W$. In addition, the reinforced kraft-paper honeycomb with a thick core and addition of flax fibres to the cementitious boards possesses better thermal conductivity, with a reduction of 42%, a lower density, and a lower water vapour transmission in comparison to the thin kraft-paper honeycomb sandwich panel.

**Keywords:** drywall panels; kraft-paper honeycomb structures; flexural strength; thermal conductivity; water vapour transmission





## 1. Introduction

The selection of construction materials has a sizeable cradle-to-grave impact on construction projects. The research on and technological development of construction materials aim to reduce costs and improve performance and environmental sustainability. Nevertheless, certain materials such as gypsum-derived materials have remained unchanged. Gypsum has been considered a vital construction material among many designers due to its excellent availability in nature, its technological and ecological properties, and the low energy consumption of manufacturing [1]. Gypsum and gypsum-derived materials have been extensively used for various purposes, such as drywall, in building construction since the early 1900s [2,3]. Gypsum-based drywall is mainly used for interior walls and ceilings in combination with either Light Steel Framing (LSF) or Light Timber Framing (LTF). A typical wall assembly consists of steel or wood studs with one or two layers of gypsum boards fixed to the stud [4].

The main advantages of gypsum-based drywall are ease of installation, low cost, fire resistance, and availability. However, gypsum-based drywall panels have disadvantages such as high thermal conductivity, low moisture and water resistance, and negative environmental impacts [5]. In this respect, drywall is not best suited for all building applications

because it does not offer any notable insulating improvements to the wall assemblies, and it could adversely affect its physical properties [5]. In addition, because drywall is easily compromised by mold and mildew growth in humid climates, in the event of a leak or flood, the lack of adequate moisture and water resistance can negatively affect the indoor air quality of the affected rooms in the building [5]. The environmental disadvantages of drywall panels are related to the insufficient facilities to recycle gypsum-based drywalls. The panels end up in landfills where they disrupt the lifecycle of anaerobic "microorganisms". Moreover, drywall panels contaminate both surface and groundwater by increasing the sulfate content in the leachate created at landfills [6]. Therefore, advancements in drywall technology should target replacing gypsum-based drywall to improve the physical and thermal properties.

Previous studies have shown that gypsum-based drywall's physical properties can be enhanced through various techniques. Adding porous materials such as expanded vermiculite (EV), expanded perlite (EP), and carbon nanomaterials (C-300, C-500, and C-750) to the gypsum-based drywall is effective in regard to improving thermal resistance by decreasing thermal conductivity [7–9]. For example, the thermal conductivity of specimens with porous materials of EV, EP, C-300, C-500, and C-750 was decreased by 30%, 20%, 35%, 36%, and 44%, respectively, compared to the samples without the addition of porous materials [8]. Furthermore, previous findings indicated that diatomite in the gypsum mix, which has a low density and porous structure, could reduce the thermal conductivity values of gypsum composites. Adding diatomite to the mixture presented an increase in porosity, resulting in a decrease in density and in the coefficient of thermal conductivity. In other words, the thermal insulation performance of composites increased by 63.8%, with the lowest thermal conductivity of 0.497 W/m·K.

Regarding water vapour permeability, gypsum boards mixed with EV and EP improved water resistance compared to gypsum boards with carbon nanomaterials [9]. Furthermore, an alternative plasterboard composed of hemp shiv bonding by lime to the conventional gypsum-based drywall showed a better performance of up to five times with respect to moisture-buffering properties than the gypsum-based drywall [10]. Although the hygrothermal behavior was improved, this study revealed that the plasterboard showed lower mechanical properties compared to the drywall [11]. Therefore, an integrated approach with other innovative materials that can improve both mechanical and hygrothermal characteristics should be considered.

Sandwich structures have become significantly popular among all the possible design ideas in composite structures due to their high performance, stiffness-to-weight ratio, and energy efficiency [12]. Typical sandwich panels are made of a core layer bonded with two face sheets [13]. While the skins are solid materials, the core can be in the form of continuous geometry, such as metallic foam or a discretized periodic geometry, e.g., honeycomb or corrugated cores. Sandwich structures have attracted the attention of industries and researchers primarily because of their higher thermal and acoustical properties [14]. Due to these advantages, sandwich structures are actively used in various engineering applications, such as civil, marine, automotive, and civil industries—particularly in building construction of roofs and internal walls [15,16].

Many sandwich panels' cores are made from rigid foam plastics such as polyurethane, polyisocyanurate, and polystyrene because of the low thermal conductivity, high moisture resistance, and low cost [15]. However, foam plastics are frequently considered hazardous fire material, which can delaminate and produce large amounts of smoke, heat, and toxic gases [16]. The sandwich structures' fire performance can be enhanced by considering an adequate core material and sufficiently restraining the facings [16]. Therefore, it can be concluded that the fundamental relationship between the structural and material parameters and the overall performance of the panel is crucial when developing a structured sandwich panel.

This research study aimed to develop an alternative kraft-paper honeycomb sandwich panel that is made of high-performance materials to replace gypsum-based drywall. The

honeycomb sandwich panel is composed of two high-density external layers bonded to an internal core layer made of a low-density honeycomb material [17,18]. The current honeycomb sandwich panels used in construction mainly focus on using non-cementitious materials for sheathing for the honeycomb structure, such as ceramics, aluminum, and wood. Distinctively, this study sheathed the core on the external faces with a cured, cementitious panel, taking a fire-resistant construction approach enforced in the building industry. The cementitious mix design consists of widely available Portland Cement (GU) and flax fibres to improve tensile strength. Natural fibres have become one of the most widely used reinforcing materials because of their sustainability, biodegradability, nontoxicity, and environmental friendliness [19]. In addition, the use of plant fibres as a tensile reinforcement for a cement matrix has recently received a lot of attention due to their lower density, better thermal insulation, high specific mechanical properties, and lower prices [19]. Moreover, the specific mechanical capabilities of flax fibres, including their stiffness, can be considered another advantage when comparing them to other natural fibres [20].

As the application's focus for this material is the residential and commercial interiors, the mix design aimed to have low water permeability and higher strength than gypsum-based drywall while remaining highly accessible worldwide. A set of experiments on this material were performed to estimate the thermal performance, flexural strength, water vapour transmission, and the density of the sandwich panel. Consequently, this research provides valuable information about the physical and mechanical properties of the new sandwich panel that are required for material characterization and modeling.

## 2. Materials and Methods

### 2.1. Materials

The cementitious mortar used for the sheathing of the sandwich panel was cast from general-use (GU) Portland Cement. Table 1 shows the typical compound compositions of Portland Cement. The oilseed flax fibres which were used for this mix design are sourced from a company based in Saskatchewan, Canada, known as Biolin Research Inc. The added flax fibres have a tensile strength between 600 and 2000 MPa and density with a range of 1.53 to 3.2 g/cm$^3$. The addition of flax fibres to the cementitious panels is effective in enhancing the crack resistance by increasing the rigidity of the samples. For the honeycomb core, two thickness alternatives of the kraft-paper-based material were used: 10.16 mm (0.4″) and 19.05 mm (0.75″). The thickness range was selected to promote future widespread adoption coming from traditional 12.70 mm (0.5″) drywall. Figures 1 and 2 show the schematic view and the laboratory view of the honeycomb sandwich structure, respectively. The honeycomb core is attached to the cementitious specimen by using an all-purpose multi-usage silicon-based adhesive from Henkel Canada Corporation that also serves as a barrier for moisture to prolong the design life of the honeycomb core material. Based on data provided by the Tricel company, the source of the kraft-paper honeycomb, the paper utilized for manufacturing the honeycomb structure is 85% recycled and 100% recyclable at the end of its design life [18].

**Table 1.** Composition of Portland Cement.

| Compound | Formula | % by Weight |
|---|---|---|
| Tricalcium silicate | $C_3S$ | 55% |
| Dicalcium silicate | $C_2S$ | 19% |
| Tricalcium aluminate | $C_3A$ | 10% |
| Tetracalcium aluminoferrite | $C_4AF$ | 7% |

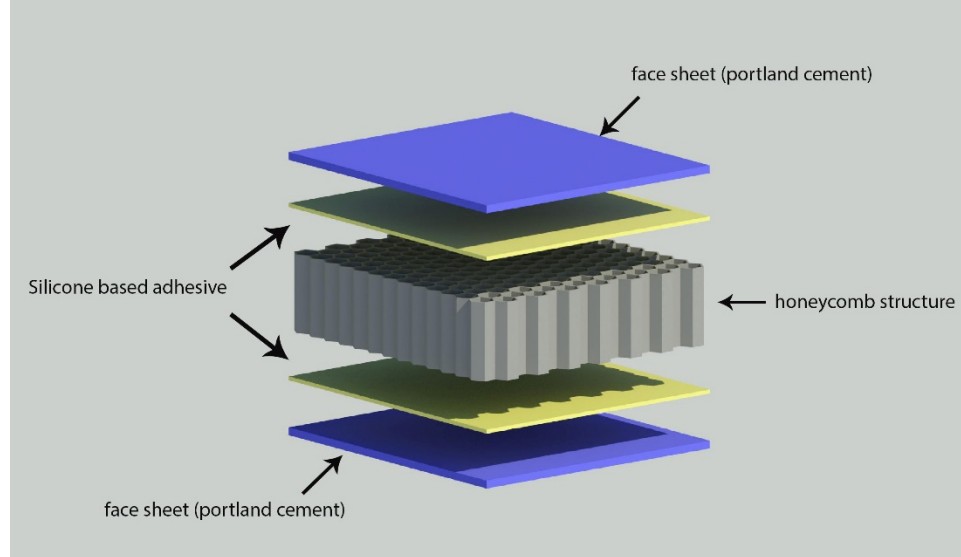

**Figure 1.** The schematic view of the honeycomb sandwich structure.

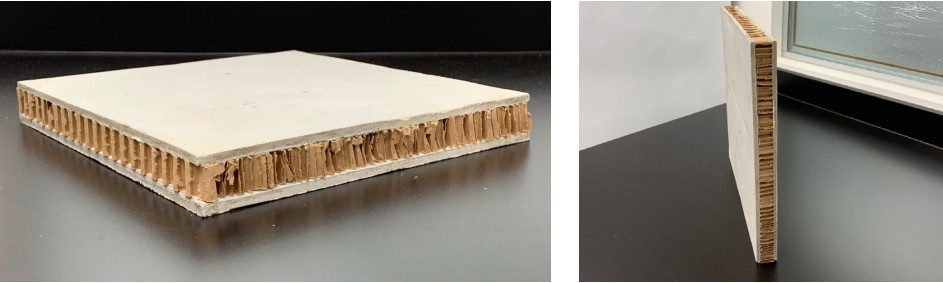

**Figure 2.** The laboratory view of the honeycomb sandwich structure.

### 2.2. Sample Preparation

The preparation of the cementitious portion of the honeycomb sandwich panels closely follows ASTM305-14 "Standard Practice for Mechanical Mixing of Hydraulic Cement Pastes and Mortars of Plastic Consistency", with minor adjustments for the addition of flax fibres [21]. Firstly, we cut flax fibres to 10 mm in length to improve their dispersion in the cement paste, and then we added them 1% by weight to the mixing water, allowing the fibres to disperse before adding the liquid to the cement powder [20]. Because the moisture absorption behaviour of the composites is influenced by the fibre content [22], 1% by weight of flax fibres was considered in order to achieve the reinforcement, while preserving thermal characteristics. Then, using a 0.45 water–cement ($w/c$) ratio, Portland Cement powder was added without mixing for 30 s to allow for the absorption as described by ASTM [21]. Next, the cement paste was mixed at a low speed for 30 s, followed by a stoppage to scrape the bowl/mixer and finishing with additional mixing for 60 s at medium speed.

The mixture was then poured into molds of two different sizes and two different shapes of rectangular and square for testing: 160 mm × 40 mm × 4 mm for flexural and 300 mm × 300 mm × 4 mm for thermal tests, according to ASTM testing procedures for thermal and flexural samples [23]. Air bubbles should be removed to achieve uniformity in the material; thus, prodding along the edges of the mold, in combination with cyclic agitation, is used to remove air bubbles. The samples were then covered in plastic and cured at room temperature (22 ± 1 °C) for 24 h before being de-molded. De-molded samples were left at room temperature for 28 days. The kraft-paper honeycomb material was then cut into two sizes to match the footprint of the cementitious flexural and thermal samples. A caulking gun was used to apply the adhesive to the interior face before evenly

distributing it with a spatula to achieve a uniform spread, 1 mm in thickness. Pressing the honeycomb material into the adhesive on both sides completes the honeycomb sandwich panel construction. The samples were cured with uniform pressure under a 1 kg weight for 72 h to ensure even bonding of the adhesive over the sample's surface area. Table 2 summarizes the composition of five types of mixed designs used for testing.

**Table 2.** Composition of kraft-paper honeycomb samples.

| Name | Mix Design | Kraft-Paper Honeycomb Thickness (mm) | Flax Fibre (%) |
|------|-----------|-------------------------------------|----------------|
| $PC_{thin}$ | Standard Portland Cement | 10.16 | - |
| $PC_{thick}$ | Standard Portland Cement | 19.05 | - |
| $Flax_{thin}$ | Flax-reinforced Portland Cement | 10.16 | 1 |
| $Flax_{thick}$ | Flax-reinforced Portland Cement | 19.05 | 1 |

### 2.3. Flexural Analysis

A ZwickRoell universal testing machine was used to determine the flexural strength of the panels according to ASTM C393 "Standard Test Method for Core Shear Properties of Sandwich Constructions by Beam Flexure" [23]. The flexure test is effective in determining the sandwich flexural stiffness and the core shear strength. Determining the mean flexural strength included testing five honeycomb sandwich panels of each category of $PC_{thin}$, $PC_{thick}$, $Flax_{thin}$, and $Flax_{thick}$ under 3-point bending. Figure 3 shows the flexural-strength-testing setup of the honeycomb sandwich panels.

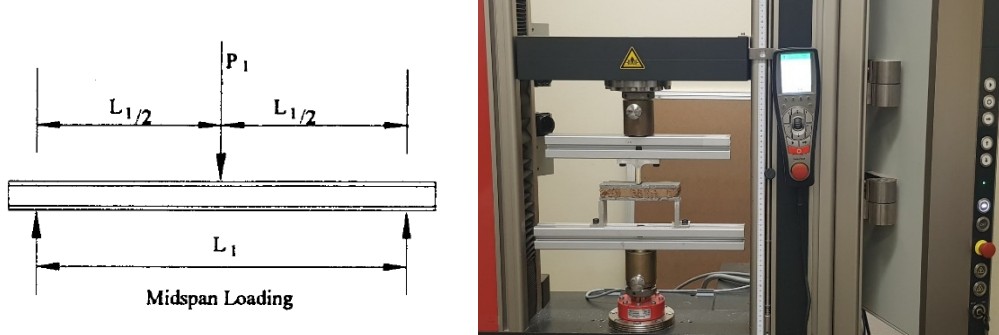

**Figure 3.** The flexural-strength-testing setup of the honeycomb sandwich panels.

The flexural strength properties of sandwich constructions were calculated according to the following Equation (1) [24]:

$$\tau = \frac{P}{(d + C)b} \tag{1}$$

where $\tau$ is the flexural strength of the sandwich panel (MPa), $P$ is the load (N), $d$ is the sandwich thickness (mm), $C$ is the core thickness (mm), and $b$ is the sandwich width (mm).

### 2.4. Thermal Analysis

The heat flow meter, located in an environmentally controlled chamber, is an accurate and widely used procedure for determining the thermal conductivity of large samples under moderate-temperature conditions [25]. Two isothermal plate assemblies and one or more heat-flux transducers are the main components of the heat-flow-meter instrument. In this research, the thermal analysis of the sandwich panels was determined by a TA Instruments Fox 314 Heat Flow Meter Apparatus (HFMA) according to international standards of ASTM C518 "Standard Test Method for Steady-State Thermal Transmission

Properties by means of the Heat Flow Meter Apparatus" [26]. This instrument is accurate for temperatures ranging from −20 to 75 °C [26]. Figure 4 shows developed samples for the thermal conductivity test.

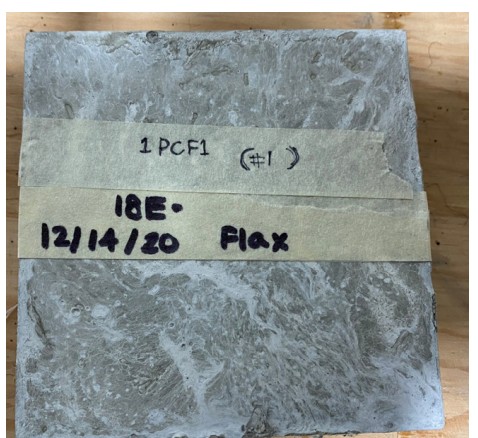 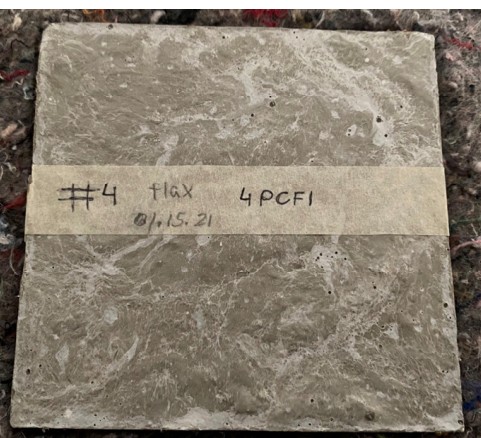

**Figure 4.** Kraft-paper honeycomb sandwich samples for thermal conductivity tests.

The specimens were positioned between two parallel plates equipped with heating and cooling systems [26] to provide steady-state and one-dimensional heat flux through the samples by setting plates at constant but different temperatures with an accuracy of ±0.01 °C [24]. The thermal conductivity was gauged given the sample thickness as the instrument fluctuated through a range of temperatures [27]. The thermal conductivity of a material in the heat flow meter method is calculated by applying Fourier's law of heat conduction with absolute thermal conductivity accuracy of ±1% at a mean temperature of plates based on the following Equation (2) [28]:

$$K = \frac{Q \times L}{A \times \Delta T} \tag{2}$$

where $K$ is the thermal conductivity coefficient of the specimen (W/m·K), $Q$ is the heat flow rate (W), $L$ is the distance between two isothermal planes (m), $A$ is the surface area of the sample (m$^2$), and $\Delta T$ is the temperature difference between upper and lower plates (K) [28].

In this study, two samples of kraft-paper honeycomb sandwich panels with different thicknesses were tested to calculate the average thermal conductivity coefficient at different temperatures.

*2.5. Density Measurement*

ASTM C271 "Standard Test Method for Density of Sandwich Core Materials" was used to determine the density of the sandwich-panel core materials [29]. The dimensions were found by using a Vernier caliper, and a digital scale was used to weigh the samples (in grams), with an accuracy of ±0.01 g. To determine the volume of the specimens, the plan dimensions and thickness of the samples were measured in millimeters. In this study, different samples of each mix design (PC$_{thin}$, PC$_{thick}$, Flax$_{thin}$, and Flax$_{thick}$) were tested to evaluate the average density of each group. The density of sandwich core materials was calculated as follows (3):

$$d = \frac{1,000,000\, w}{v} \tag{3}$$

where $d$ is the density (kg/m$^3$), $w$ is the final mass (g), and $v$ is the final volume (mm$^3$).

*2.6. Water Vapour Transmission*

Several experiments on each sample type were carried out according to the ASTM E96/E96M "Standard Test Methods for Water Vapour Transmission Materials" to measure

water vapour transmission through permeable and semipermeable materials [30]. Among the applicable methods, the water method was considered for the measurement of permeance [31]. Figure 5 shows the water-vapour-transmission sample developed according to the water method.

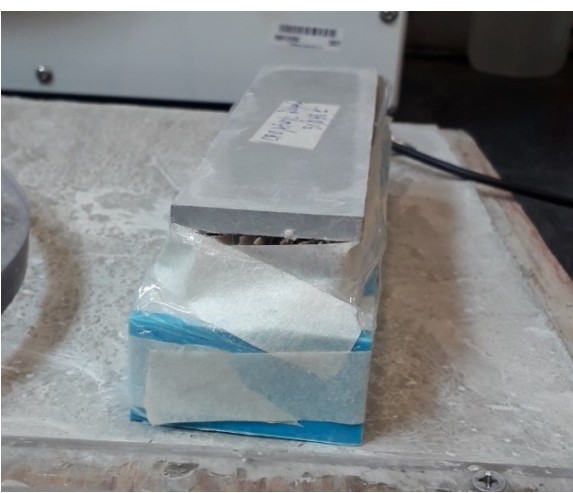

**Figure 5.** Water-vapor-transmission sample.

In the water method, the dish impermeable to water or water vapour is filled with distilled water to a level of [19 mm $\pm$ 6 mm] from the sample throughout the experiment. Each of the samples is attached to a dish with a sealant resistant to the passage of water vapour. The dish assembly was placed in a chamber with a controlled temperature of 23 °C and relative humidity of 50% and then weighed every 24 h. The measure of vapour transfer through the sample from the water to the controlled atmosphere can be determined by weighing the sample [32]. After measuring the mass change, the water vapour transmission of the samples is calculated as follows (4):

$$WVT = \frac{G}{tA} = \left(\frac{G}{t}\right)A \tag{4}$$

where $WVT$ is the rate of water vapour transmission (g/h·m$^2$), $G$ is weight change (from the straight line) (g), $t$ is the time (h), $G/t$ is the slope of the straight line (g/h), and $A$ is the test area (cup mouth area) (m$^2$).

Water vapour permeance, which is the water vapour transmission rate of a membrane influenced by the unit vapour pressure difference through a unit thickness, can be quantified as follows (5) [32]:

$$Permeance = \frac{WVT}{\Delta P} = \frac{WVT}{S(R_1 - R_2)} \tag{5}$$

where $\Delta P$ is vapour pressure difference (mm Hg, $1.333 \times 10^2$ Pa), $S$ is the saturation vapour pressure at the test temperature (mm Hg, $1.333 \times 10^2$ Pa), $R_1$ is the relative humidity at the source expressed as a fraction (in the dish for water method), and $R_2$ is the relative humidity at the vapour sink expressed as a fraction.

## 3. Results

### 3.1. Flexural Analysis

Figure 6 shows the results obtained from the flexural analysis. A force-displacement graph is displayed to assess the behavior of each sample type, namely PC$_{thin}$, PC$_{thick}$, Flax$_{thin}$, and Flax$_{thick}$, under three-point bending.

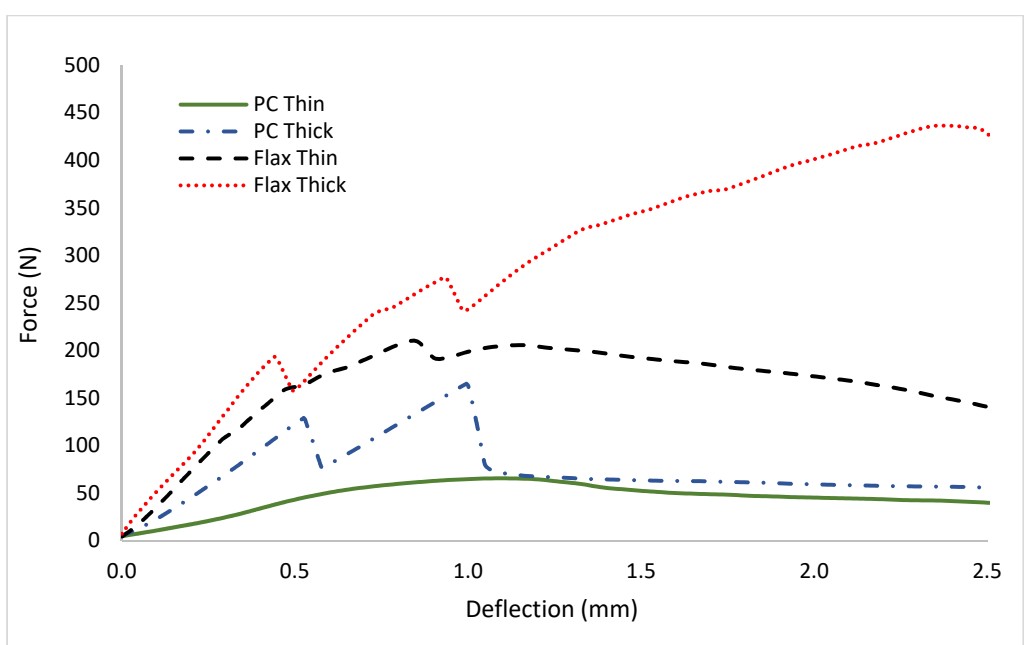

**Figure 6.** Force displacement graph of honeycomb sandwich panels under 3-point bending. (Note: Only 1 test for each sample is shown based on the avg. peak strength.)

The force-deflection graph for three-point bending indicates a specimen's ability to withstand a force until it reaches the point of failure, which is the point where the material can no longer support an increase in load. Abrupt changes in the force of all sample types in the graph signify cracking in the Portland Cement sheathing. The findings indicated that $PC_{thin}$ and $PC_{thick}$ exhibited early stage cracking due to the brittle nature of the sheathing. The Portland Cement samples cracked during the gluing and testing stages due to the lack of tensile reinforcement. Furthermore, as the honeycomb sandwich panels are composite materials, the deformation of this material involves both elastic and plastic behavior.

The addition of flax fibre may cause a lack of uniformity throughout samples due to the tendency of fibres to "clump". Nevertheless, the results showed that adding flax fibre increased the "sample" stiffness compared to the non-flax samples based on the slight increase in the slope of the graph. The elastic modulus and strength of a composite can be predicted by using the rule of mixture equation, $E_c = fE_f + (1 - f)E_m$, based on the E modulus of the matrix $(E_m)$ approximately between 10 and 40 MPa, E modulus of fibres $(E_f)$ ranging from 12 to 35 GPa, and volume fraction of fibres $(f)$ [33]. Therefore, the addition of flax fibre, which has a higher elastic modulus, increased the samples' elastic modulus.

Before the failure, the deflection limit was similar among Portland Cement samples, which consistently withstand more significant displacement before collapse than the flax samples, but only at less sustained forces.

Additionally, initial cracks in the samples were prolonged and occurred at higher forces, especially for the $Flax_{thick}$ sample. All specimens showed decreased stiffness after cracking, as the force becomes more reliant on the kraft-paper honeycomb. The curved nature of each graph before the failure point indicated its plastic deformation, in which the honeycomb appears to withstand most of the force due to the two or three continuous sheath cracks in each specimen.

As all samples have a uniform span, normalizing the equation changes the values over the *x*-axis consistently. The findings demonstrated that honeycomb with a thickness of 19.05 mm had a better flexural strength than the one with a thickness of 10.16 mm, as it can tolerate more loads. Moreover, the addition of flax fibres is effective in prolonging early stage cracking due to the tensile characteristics they add to the honeycomb sandwich panels.

Table 3 and Figure 7 show the average ultimate strength of each sample type and the standard deviation bars. Based on the findings, $PC_{thin}$ had the lowest average maximum

strength, i.e., 0.87 MPa. However, the average ultimate strength was slightly higher, at 1.02 MPa. Contrarily, both Flax$_{thin}$ and Flax$_{thick}$ exhibited consistent, reliable results with lower standard deviations. The change in honeycomb thickness from 10.16 to 19.05 mm for the flax samples resulted in a considerable increase in the average ultimate strength, from 1.55 to 2.11 MPa. It was verified that the principal factors in determining the sample's stiffness are the material's modulus of elasticity and the structure's geometry in terms of the planar moment of inertia. Therefore, thick samples in which layers are located further from the axis of motion resulted in higher stiffness. Thus, the sandwich panel could be more resistant due to the higher stiffness. In addition, as the maximal strength is dependent on the combination of sandwich thickness and the core thickness, the thicker core withstood more loads than the thinner one. Overall, it was determined that the samples with flax fibres exhibited less variability in results, and their flexural strength is superior to the PC panels. By using the average values, we determined that Flax$_{thin}$ performed 44% better than the PC$_{thin}$, and Flax$_{thick}$ performed 52% better than the PC$_{thick}$. In addition, as flax specimens showed higher strength compared to the gypsum board (12.7 mm thickness), with a strength of 1.8 MPa [34], this study was mainly focused on analyzing the thermal performance, density, and water vapour transmission of flax fibre samples.

**Table 3.** Honeycomb-sandwich-panel ultimate strength.

| Sample | Average Strength (MPa) | Standard Deviation (MPa) |
|---|---|---|
| PC$_{thin}$ | 0.87 | 0.40 |
| PC$_{thick}$ | 1.02 | 0.31 |
| Flax$_{thin}$ | 1.55 | 0.13 |
| Flax$_{thick}$ | 2.11 | 0.14 |
| Gypsum Board | 1.8 | 0.45 |

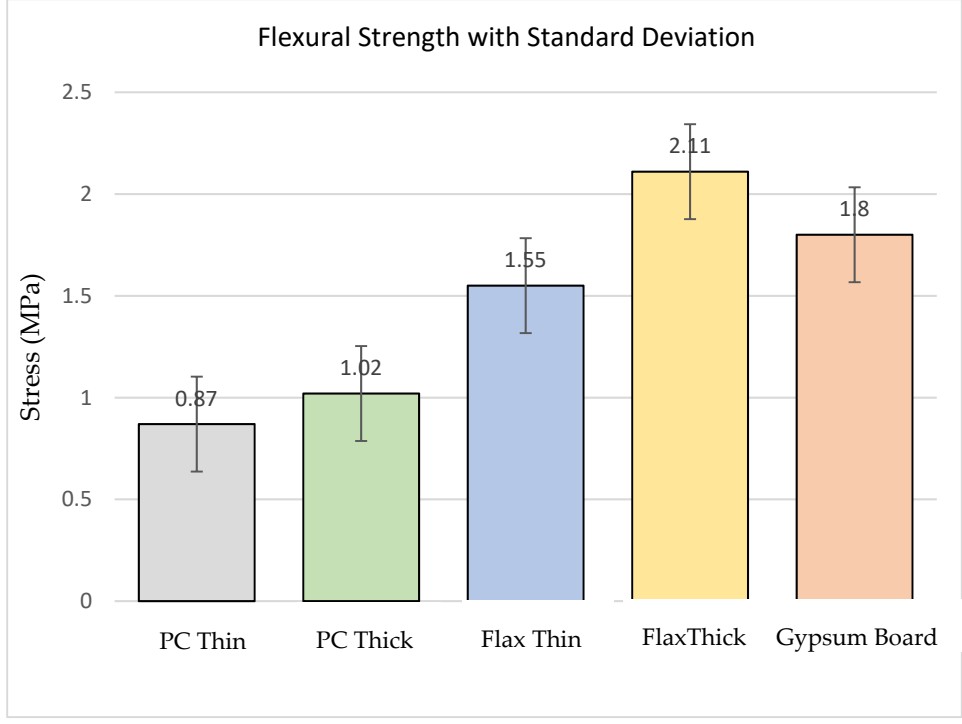

**Figure 7.** The average ultimate strength of honeycomb sandwich panels with standard deviation.

*3.2. Thermal Analysis*

This section presents and discusses the thermal performance of two different thicknesses of honeycomb sandwich panels with the addition of flax fibres. Figure 8 shows the

average thermal conductivities of the sandwich panels and the gypsum board. Overall, there is a significant difference between the thermal conductivity curves of the gypsum board and sandwich samples. The findings indicated that the thermal conductivity of kraft-paper honeycomb sandwich panels was lower than that of the gypsum board. The incorporation of kraft-paper honeycomb in the cementitious panels has an interrelated effect on decreasing the thermal conductivity of samples because of the lower thermal conductivity of kraft-paper honeycomb compared to gypsum board. The results showed that the average thermal conductivities of gypsum board, $Flax_{thin}$, and $Flax_{thick}$ were 0.159, 0.097, and 0.092 W/m·K, respectively. It was confirmed that the thick sample showed a larger decrease in the thermal conductivity compared to the thinner specimen. The likely reason is that, by increasing the core thickness, the amount of air volume in the volume of the honeycomb increases. Therefore, a decrease in the sample's thermal conductivity happened due to the low thermal conductivity of air compared to the thermal conductivity of the kraft-paper honeycomb sandwich panel. The thermal conductivity of the gypsum board ranges from 0.154 to 0.164 W/m·K, with an average of 0.159 W/m·K. However, the thermal conductivity of $Flax_{thin}$ and $Flax_{thick}$ was reduced by 39% and 42%, respectively, compared to the gypsum board.

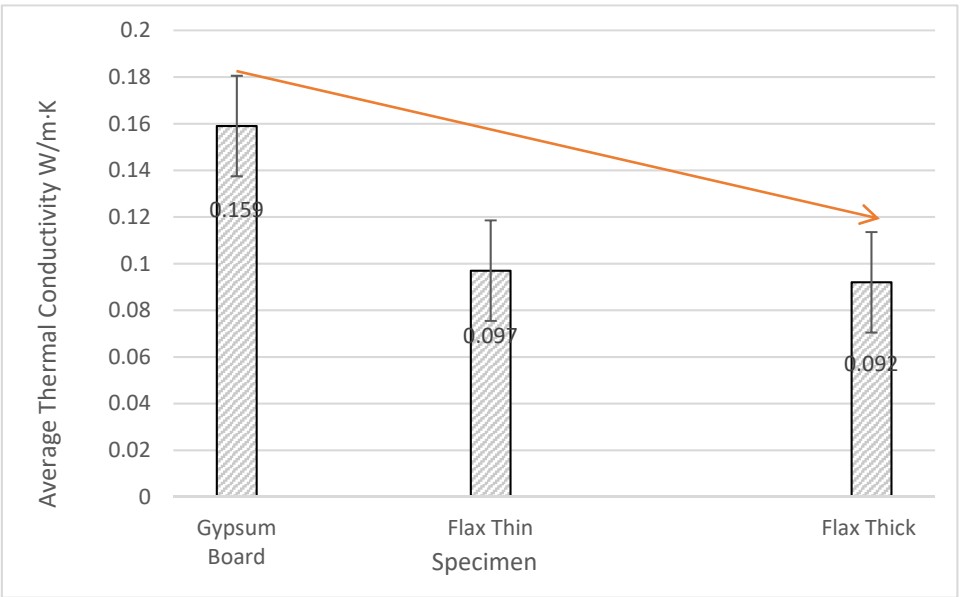

**Figure 8.** Average thermal conductivity of gypsum board, $Flax_{thin}$ and $Flax_{thick}$.

　　　Figure 9 compares the mean value of the coefficient of thermal conductivity of two kraft-paper honeycomb sandwich panels measured as a function of temperature, ranging from −10 to 80 °C, against the thermal conductivity of a gypsum board. The results illustrate that the thermal conductivity of both samples increased almost linearly with the increasing temperature. However, the coefficients of thermal conductivity of both $Flax_{thin}$ and $Flax_{thick}$ were lower as compared to the gypsum board. This is due to the absence of cracks in the flax-fibre-reinforced panels, which decreased the rate of heat transfer in the specimens. In addition, as natural fibres contain microporosity, the panels that contain these fibres can effectively reduce the heat conductivity of the specimens. Moreover, previous studies have shown that incorporating natural fibres into cement-based composites could have effects on improving their thermal performance [35]. In reality, the addition of fibres to a material result in higher porosity, which can reduce its thermal conductivity [36]. In a kraft-paper honeycomb sandwich panel, heat is transferred by conduction through the core and paper and convection in the air voids. It is noticeable that, as the sample becomes thicker, a higher decrease in conduction occurs due to the inverse proportion of the rate of heat transfer to the thickness of the sample. Therefore,

as depicted in Figure 8, Flax$_{thick}$ exhibited a lower coefficient of thermal conductivity when compared to its thin counterpart. In this respect, the thermal conductivity of the thick sample ranges from 0.080 to 0.107 W/m·K, with an average value of 0.092 W/m·K. However, the thermal conductivity of the thin sample is approximately 5.4% higher, ranging from 0.084 to 0.117 W/m·K, with an average of 0.097 W/m·K. Therefore, as Flax$_{thick}$ has lower thermal conductivity in comparison to Flax$_{thin}$ and gypsum board, it shows better thermal performance and can be considered a good thermal insulator.

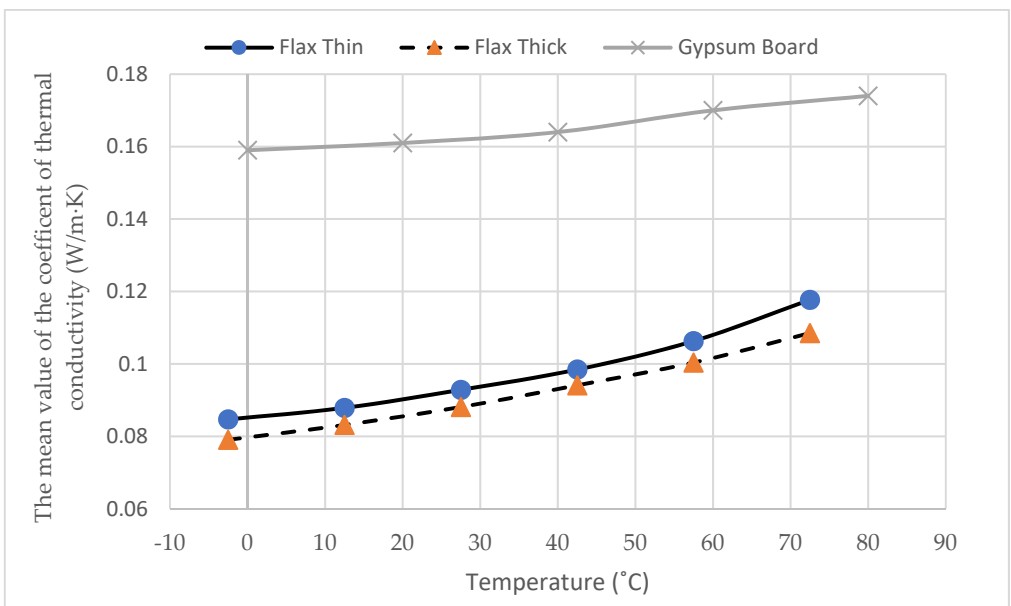

**Figure 9.** Thermal conductivity of kraft-paper honeycomb sandwich panels compared to gypsum board.

### 3.3. Density Measurement

Table 4 summarizes the kraft-paper honeycomb cementitious panels' volume, mass, standard deviation, and samples' densities. A low-density material is favorable because, historically, it can provide thermal insulation. As density is a measure of mass per volume, the average density of a specimen equals its total mass divided by its total volume. Figure 10 presents the samples within the density range from 0.60 to 0.10 g/cm$^3$. Possibilities for differences in the density of samples include different values of mass and volume due to the different honeycomb core thicknesses. In addition, samples with different thicknesses resulted in different densities because the relative density of honeycomb sandwich panels is proportional to the wall thickness to a wall-length ratio (t/L). Moreover, the gypsum board with a density of 0.65 g/cm$^3$ presented a lower value due to its porous structure compared to the sandwich panels [1].

**Table 4.** Honeycomb-sandwich-panel densities.

| Sample | Average Volume (mm$^3$) | Average Density (g/cm$^3$) | Standard Deviation (g/cm$^3$) |
|---|---|---|---|
| PC$_{thin}$ | 178,360 | 0.99 | 0.085 |
| PC$_{thick}$ | 251,400 | 0.76 | 0.069 |
| Flax$_{thin}$ | 250,000 | 0.89 | 0.068 |
| Flax$_{thick}$ | 270,500 | 0.67 | 0.051 |

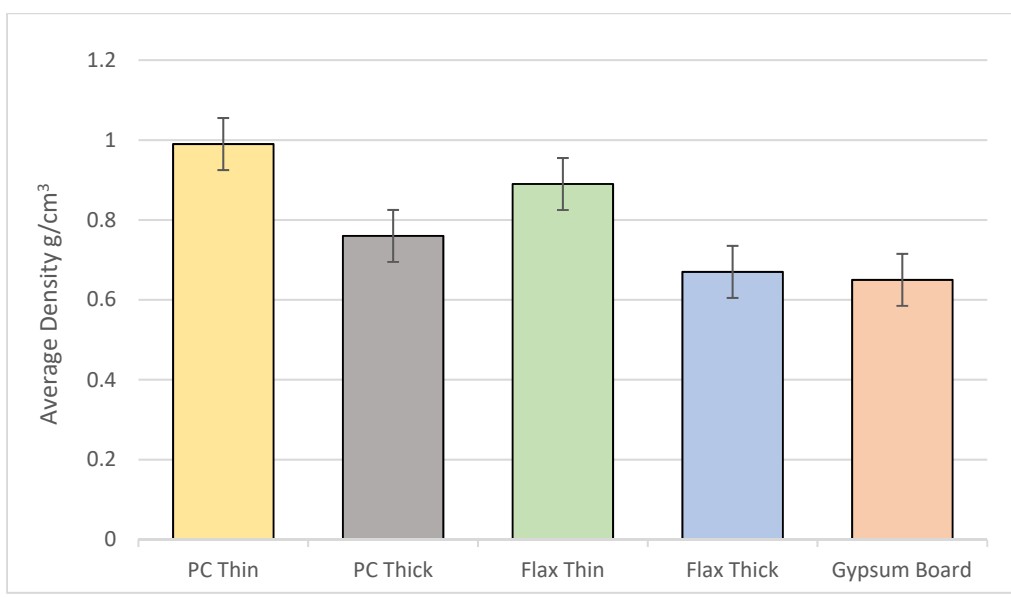

**Figure 10.** Average densities of honeycomb structures.

In the category of Portland Cement samples, $PC_{thick}$ has the lower average density and standard deviation in comparison to $PC_{thin}$. On the other hand, the average density and standard deviation of $Flax_{thick}$ is lower compared to $Flax_{thin}$. Therefore, the results revealed that the thick specimens of the study had the lowest densities, with 0.76 and 0.67 $g/cm^3$. The likely reason for the decrease in $Flax_{thick}$ is the more significant amount of void volume than $Flax_{thin}$. Although $Flax_{thin}$ and $Flax_{thick}$ share the same cross-sectional composition, $Flax_{thick}$ has a larger height, making the void volume much more significant compared to $Flax_{thin}$. Therefore, a thicker honeycomb core results in a lower density compared to the thinner specimens. In addition, the other explanation is related to the highest average volume of approximately 270,500 and 251,400 $mm^3$ of $Flax_{thick}$ and $PC_{thick}$, thus resulting in lower densities. However, $PC_{thin}$ and $Flax_{thin}$ with average volumes of 178,360 and 250,000 $mm^3$ have the highest average densities, i.e., 0.99 and 0.89 $g/cm^3$, respectively.

*3.4. Water Vapour Transmission*

Water vapour transmission indicates the degree to which water vapour is transferred through a substance under specific temperature conditions and from regions of high relative humidity to regions of low relative humidity. The findings indicated a linear relationship between the total weight loss of dish assembly of water as a function of time. According to the ASTM standard, the slope of the line corresponds to the term of water vapour transmission rate. Figure 11 illustrates the average values of water vapour transmission of the flax-reinforced samples after reaching a steady state at the time of 23 h, where the rate of weight change is substantially constant. The results showed that $Flax_{thick}$ had a lower average water vapor transmission, with the amount of 9.570 $g/h·m^2$, compared to $Flax_{thin}$, with the amount of 17.195 $g/h·m^2$. It was confirmed that water vapour transfer through building components is mainly dependent on two mechanisms: air leakage and diffusion [37]. However, as a kraft-paper honeycomb is a closed-cell structure, air leakage can be considered less of a significant problem than diffusion. In addition, as water vapour transmission is affected by the thickness of the materials, $Flax_{thick}$ exhibited a higher decrease in the rate of water vapour diffusion. Figure 12 compares the average water permeance of two kraft-paper honeycomb sandwich panels and the gypsum board. Water vapour permeance is defined as "*the timed rate of water vapour transmission through unit area of flat material or construction induced by a unit vapour pressure difference between two specified surfaces, under specified temperature and humidity conditions*" [32]. A hygroscopic material has a great ability to absorb and/or desorb water from the surrounding environment until it reaches an equilibrium condition [38]. Gypsum board is a hygroscopic material,

and as such, the results indicated that the gypsum board's permeance is much higher compared to the developed sandwich panels. As permeance is inversely proportional to the thickness and directly related to water vapour transmission rate, Flax$_{thick}$, with a rise of core thickness, resulted in lower permeance rather than Flax$_{thin}$. The average permeance of Flax$_{thin}$ and Flax$_{thick}$ was 1.398 perms and 1.338 perms, respectively. Therefore, Flax$_{thick}$ can be considered an adequate water-resistant material.

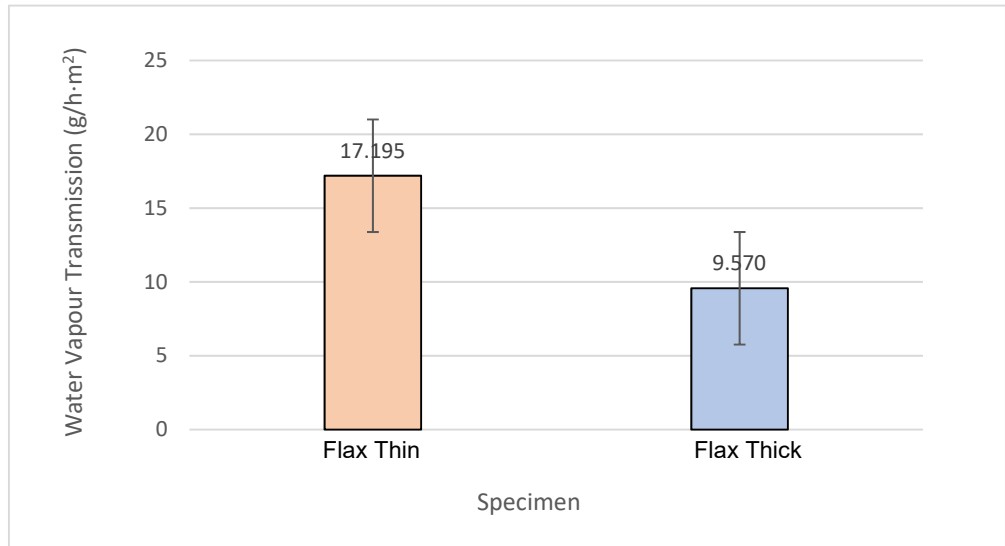

**Figure 11.** Average water vapour transmission of honeycomb structures.

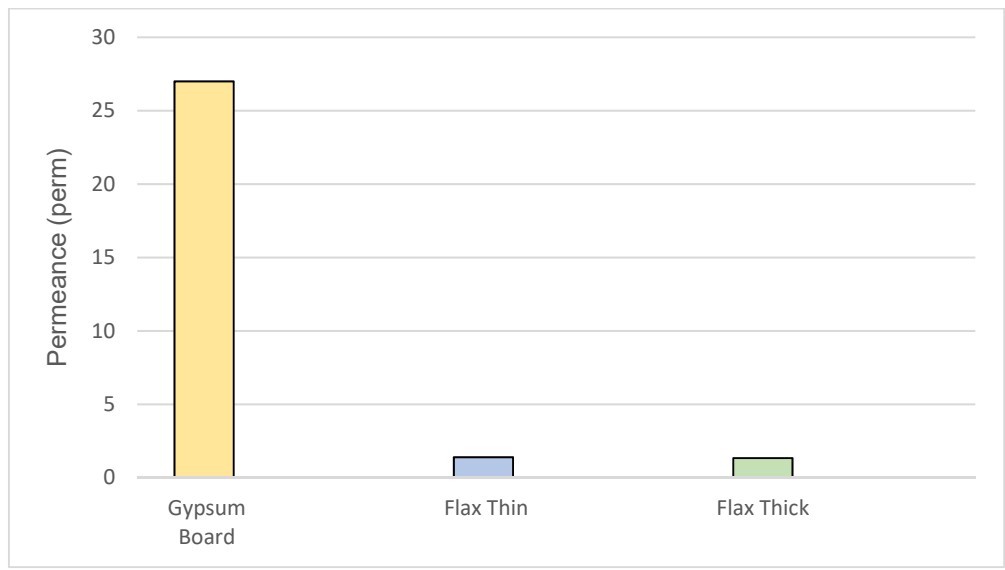

**Figure 12.** Average permeance of honeycomb structures.

## 4. Conclusions

This study developed kraft-paper honeycomb sandwich panels consisting of two different cementitious boards (Portland Cement and Portland Cement reinforced with flax fibres) to improve the mechanical and physical properties of drywall panels as an alternative to the gypsum-based boards in residential and commercial buildings. In addition, the results and findings of this report provide new and valuable design recommendations regarding the material properties and the temperature range of new kraft-paper honeycomb sandwich panels. The main conclusions achieved in this investigation can be summarized as follows:

- The incorporation of flax fibres into the Portland Cement mixtures increases samples' stiffness and prolongs the early stage cracking of non-flax-fibre specimens. These significant enhancements are because of the unique properties of flax fibres in resisting greater bending and fracture forces than the brittle Portland Cement mixtures. Particularly, the 19.05 mm kraft-paper honeycomb structure has approximately 41% higher flexural performance than the kraft-paper honeycomb sandwich panel with a thickness of 10.16 mm. As both the sandwich thickness and core thickness factors influence the maximal strength, the sample with a thicker core tolerated higher loads better than a thin sample. In addition, increasing the core thickness resulted in higher stiffness and resistance against bending loads due to the higher moment of inertia.
- Due to flax fibres' lower thermal conductivity and better thermal insulation properties, all flax-fibre-reinforced panels exhibited lower thermal conductivity than the gypsum-based drywall. However, the thermal conductivity of $Flax_{thick}$ showed a greater reduction, e.g., by 42%, compared to the gypsum board. The thermal analysis results confirmed that the samples' air volume increased by raising the thickness of the sample. Therefore, $Flax_{thick}$ effectively decreased the coefficient of thermal conductivity compared to $Flax_{thin}$ in different temperatures and is more resistant against a heat flow due to its higher embodied porosity. The results indicated that the thermal conductivity of $Flax_{thin}$ as a function of temperature, ranging from $-10$ to 80 °C, is relatively 5.4% higher than the thermal conductivity of $Flax_{thick}$
- The density measurements carried out on kraft-paper honeycomb sandwich panels showed a reduction in density by adding flax fibres to the cementitious boards. In addition, for equal fibre content, the density of the sample with a thick honeycomb core is approximately 24% lower than that of $Flax_{thin}$. This remains possible due to the highest average volume and larger void volume in $Flax_{thick}$ samples compared to the specimens with a thicker core.
- The water vapour transmission and permeance analysis indicated that $Flax_{thick}$ samples with a core thickness of 19.05 mm had lower values, with the amounts of 8.88 g/h·m$^2$ and 1.291 perms, respectively, in comparison to the $Flax_{thin}$ sample. In effect, as the kraft-paper honeycomb is a closed-cell structure, the diffusion mechanism mainly affects the rate of water vapour transmission. Therefore, samples with thick cores exhibited a greater decrease in the amount of water vapour diffusion and permeance.

Therefore, this study highlighted that the impregnation of flax fibres in a multilayer sandwich structure with a thicker core resulted in a higher flexural strength, superior thermal performance, lower density, and lower water vapour transmission rate. Therefore, flax-fibre-reinforced kraft-paper honeycomb cementitious sandwich panels can be considered a functional material for improving the physical and mechanical properties of drywall, while providing a substantial increase in R-value when compared to $Flax_{thin}$ and gypsum board. Furthermore, drywall is highly susceptible to microbial growth due to the cellulosic paper backing. This microorganism can affect the thermal comfort and air quality of a dwelling unit over time. Therefore, since the environmental impacts of mold and mildew growth are essential factors in material selection, further research is needed to evaluate the hygrothermal performance and environmental interactions of the proposed kraft-paper honeycomb sandwich panel. In this regard, subsequent work should include the use of collected experimental data as an input to a numerical computer simulation to validate the developed model due to the lack of comparison of numerical simulation results and the experimental measurements.

**Author Contributions:** Conceptualization, S.S. and R.F.; formal analysis, S.S.; funding acquisition, R.F.; investigation, S.S., N.S., M.K. and R.F.; methodology, R.F.; project administration, R.F.; resources, M.M.; supervision, M.K.; validation, M.M.; writing–original draft, S.S.; writing—review and editing, N.S. and M.K. All authors have read and agreed to the published version of the manuscript.

**Funding:** The authors gratefully acknowledge the financial support received from the Natural Sciences and Engineering Council of Canada (NSERC).

**Institutional Review Board Statement:** Not applicable.

**Informed Consent Statement:** Not applicable.

**Data Availability Statement:** Not applicable.

**Conflicts of Interest:** The authors declare no conflict of interest.

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
