# Peer review of "Cementitious Insulated Drywall Panels Reinforced with Kraft-Paper Honeycomb Structures"

_buildings, doi:10.3390/buildings12081261_

Round 1

Reviewer 1 Report

Cementitious Insulated Drywall Panels Reinforced with Kraft Paper Honeycomb Structures

This study aims to develop sandwich panels with improvements in flexural strength and thermal insulating properties 14 through the combined use of cementitious binder mix and kraft paper honeycomb structures.

Section 2.3, a schematic and a  real photo of test setup could be very useful. 

Section 2.4, figures are required as well.

How about failrue modes of all tested specimens, any description along with figures?

Authors must summarized results in more systematic way with reference to the previous studies.

Conclusiosn are too length as well, there is need to revised properly. 

Author Response

Manuscript #:  buildings-1832692

Title of MS: CEMENTITIOUS INSULATED DRYWALL PANELS REINFORCED WITH KRAFT PAPER HONEYCOMB STRUCTURES

Authors: Sepideh Shahbazi, Nicholas Singer, Muslim Majid, Miroslava.Kavgic and Reza Foruzanmehr

The paper in the present form has been revised to address the reviewer’s comments.

REVIEWER #2

Comment

Response

Section 2.3, a schematic, and a real photo of test setup could be very useful.

The authors appreciate this comment, and the photos were added.

Section 2.4, figures are required as well.

The authors appreciate this comment, and the photos were added.

How about failure modes of all tested specimens, any description along with figures?

The authors appreciate this comment, all tested specimens went under a brittle fracture.

Authors must summarize results in more systematic way with reference to the previous studies.

The authors appreciate this comment, and results were revised, and more references were added.

Conclusions are too long as well, there is need to revise properly.

The authors appreciate this comment, and the conclusions were revised.

Reviewer 2 Report

Manuscript title: Cementitious Insulated Drywall Panels Reinforced with Kraft Paper Honeycomb Structures

The paper is well-written, and the investigations reported in the paper are interesting and present a new topic related to sandwich panels. All important aspects of the results are discussed. However, minor corrections should be considered before accepting the paper for publication, according to the following suggestions:

1-      Line 19, in the abstract, please specify the strength e.g. the sample flexural strength

2-      Line 21, to what extent R-value increase.

3-      Regarding the plant fibers and other natural fibers, it is recommended to add more literature to support this. You can refer to https://www.mdpi.com/1996-1944/15/10/3601

4-      Please supply the physical and mechanical properties of the used flax fibres and kraft paper-based materials.

5-      It is recommended to provide a laboratory picture beside the schematic one.

6-      It is recommended to provide the testing set-up of the honeycomb sandwich panels.

7-      What are the specimens’ dimensions of the tested panels?

8-      Enhance the quality of all figures.

9-      Conclusions are too long. It should be focused on the focal findings of the study and can be presented in points form.

Author Response

Title of MS: CEMENTITIOUS INSULATED DRYWALL PANELS REINFORCED WITH KRAFT PAPER HONEYCOMB STRUCTURES

Authors: Sepideh Shahbazi, Nicholas Singer, Muslim Majid, Miroslava.Kavgic and Reza Foruzanmehr

The paper in the present form has been revised to address the reviewer’s comments.

REVIEWER #1

Comment

Response

I am somewhat concerned about measurement of rheology of a powder.  Yes you have diluted it in liquid, but results would be entirely dependent on concentrations - how did you choose the concentrations you used?

The authors appreciate this comment, one part was added to the Viscometry section:

Page 5:

To determine the water-to-CP ratio for a paste-like material (or slurry), downscaled Abrams cone mini slump test was used [16]. This method approximates the yield stress or the threshold of applied stress at which deformation of the material becomes plastic. Different concentration of CP was tested at an increment of 10 ml of distilled water. 60 mL of distilled water mixed with 100g of CP, was found to give the appropriate consistency of the paste, with no excess water or clumps of solid as seen in Figure 3. Above or below this ratio, the slump, which is inversely correlated to the yield stress (measured 1350 KPa in average), did not demonstrate a good workability [17].

I saw a few typos so a thorough editorial review would be good.  Beware of acronym abuse.

The authors appreciate this comment, the article was revised, and typos were corrected.

Line 19, in the abstract, please specify the strength e.g., the sample flexural strength. 

The authors appreciate this comment, the article was revised, and the specific strength was added and highlighted in the manuscript.

Line 21, to what extent R-value increases?

The authors appreciate this comment and the R-values of the materials have been added to the article and highlighted.

Regarding the plant fibres and other natural fibres, it is recommended to add more literature to support this.

The authors appreciate this comment, and more references have been added to the article in this regard.

Please supply the physical and mechanical properties of the used flax fibres and kraft paper-based materials. 

The authors appreciate this comment, and some of the properties were added to the article.

It is recommended to provide a laboratory picture beside the schematic one.

The authors appreciate this comment, and the laboratory pictures were added.

it is recommended to provide the testing set-up of the honeycomb sandwich panels. 

The authors appreciate this comment, and testing setups of the specimens were added to the article.

What are the specimens' dimensions of the tested panels? 

The authors appreciate this comment. In this article samples with width of 31 mm and span of 150 mm were considered. In terms of the honeycomb structure, two different thicknesses of honeycomb cores were attached to the cementitious panels with thickness of 6 mm.

Enhance the quality of all figures.

The authors appreciate this comment, and the figures were revised.

Conclusions are too long. It should be focused on the focal findings of the study and can be presented in points form.

The authors appreciate this comment, and the conclusions were revised.

Author Response

Manuscript #:  buildings-1832692

Title of MS: CEMENTITIOUS INSULATED DRYWALL PANELS REINFORCED WITH KRAFT PAPER HONEYCOMB STRUCTURES

Authors: Sepideh Shahbazi, Nicholas Singer, Muslim Majid, Miroslava.Kavgic and Reza Foruzanmehr

The paper in the present form has been revised to address the reviewer’s comments.

REVIEWER #3

Comment

Response

Abstract: Please re-write lines 18-19 to be clearer.

The authors appreciate this comment, the specific lines were revised.

Lines 105-108: It is possible to improve the literature by citing more recent studies on the use of flax fibers.

The authors appreciate this comment, and more references have been added to the article in this regard.

To be clearer, please put some photos from the experimental tests including flexural and thermal.

The authors appreciate this comment, and the pictures were added.

Lines 256-260: You mention the use of “rule of mixture” to calculate the elastic modulus and strength. How much is the fiber volume of flax? What about the E modulus of fibers and the matrix?

The authors appreciate this comment, and the E modulus of fibre and matrix were added to the article.

Also, in this article we have used the weight percent of fibre instead of volume percent because of the variability of density of flax fibres.

The flexural strength was gathered in Table.3 and Figure.3, while the formula given in 2.3 was for shear stress. Please explain.

The authors appreciate this comment, the forumal was revised, and typos were corrected.

Lines 326-328: It is true that the addition of natural-based fibers to cement-based materials could enhance thermal performance. To justify it you can add some citations.

The authors appreciate this comment, and more references have been added to the article in this regard.